Majority clustering for imbalanced image classification

Sharma Keshav 1
Arora Jyoti joy.arora@gmail.com 1
Kherwa Pooja 2
Alansari Zainab zainab.alansari@utas.edu.om 3
1 Information Technology, Maharaja Surajmal Institute of Technology , New Delhi , India
2 Computer Science & Engineering, Maharaja Surajmal Institute of Technology , New Delhi , India
3 University and Technology of Applied Sciences , Muscat , Oman
Bolshoy Alexander
Electronic publication date: 2025 Jun 30
Publication date: 2025
Volume: 11
Electronic Location ID: e2891
Received 2024 Oct 25; Accepted 2025 Apr 22
Copyright: ©2025 Sharma et al.
Copyright year: 2025
Copyright holder: Sharma et al.
License: This is an open access article distributed under the terms of the Creative Commons Attribution License, which permits unrestricted use, distribution, reproduction and adaptation in any medium and for any purpose provided that it is properly attributed. For attribution, the original author(s), title, publication source (PeerJ Computer Science) and either DOI or URL of the article must be cited.
License URL: https://creativecommons.org/licenses/by/4.0/

Keywords: Imbalanced datasets, Classification, K-means clustering, ResNet-18, Class imbalance

Funding: The authors received no funding for this work.

==============================
Class imbalance is a prevalent challenge in image classification tasks, where certain classes are significantly underrepresented compared to others. This imbalance often leads to biased models that perform poorly in predicting minority classes, affecting the overall performance and reliability of image classification systems. In this article, an under-sampling approach based on reducing the samples of majority class is used along with the unsupervised clustering approach for partitioning the majority class into clusters within the datasets. The proposed technique, Majority Clustering for Imbalanced Image Classification (MCIIC) improves the traditional binary classification problems by converting it into multi-class problem, thereby creating the more balanced classification solution to the problems where one need to detect rare samples present in the dataset. By utilizing the elbow method, we determine the optimal number of clusters for the majority class and assign each cluster a new class label. This complete process ensures a balanced and symmetrical class distribution, effectively addressing imbalances both between and within classes and helps to perform imbalanced classification. The effectiveness of the proposed model is evaluated on various benchmark datasets, demonstrating their ability to improve the predictive performance of the proposed MCIIC on imbalanced image datasets. Through empirical evaluation, we showcase the impact of proposed technique on model accuracy, precision, recall, and F1-score, highlighting its importance as a pre-processing step in handling imbalanced image datasets. The results highlight the significance of proposed model as a practical approach to address the challenges posed by imbalanced data distributions in machine learning tasks.

Introduction

In today’s modern world, machine learning is gaining importance as it allows the computer to learn and evolve based on existing data without being explicitly programmed. In this, realm image analysis plays an important role, as the computer learns by interpreting this visual information. People are surrounded by images and visuals everywhere. Understanding and analysing these images deeply impacts the lives of people. Various deep learning methods are employed across various domains such as health care, robotics, computer vision and natural language processing and have showcased unprecedented effectiveness. A crucial factor that affects the efficiency of their performance is the availability of the training dataset (Perez & Wang, 2017). In essence, the more diverse and abundant the training data, the better the deep learning model can learn to generalize patterns and make accurate predictions. But due to scarcity of the data and unbalanced dataset, sometimes these algorithms failed to give the desired outcomes (Razzak, Naz & Zaib, 2018).

In machine learning, the problem of imbalanced datasets is encountered very commonly. This problem occurs when the count of the instances of majority class are far more than the count of the instances of minority classes (Arora et al., 2022). The algorithms based on supervised learning are most frequently used for the process of classification are inclined towards majority class (the class with more instances), thus neglecting the minority samples. In certain cases, such as when a diagnosis of tuberculosis or cancer cells is made, the decision must be made based on the minority class, which has the fewest cases (Devarriya et al., 2020; Shen et al., 2023). Despite the remarkable performance of different classification models across diverse fields, their performance can be degraded by the presence of highly imbalanced class distributions in emerging datasets. This imbalance poses a critical challenge to the model’s ability to generalize effectively (Zhu et al., 2020). In such cases the model may perform poorly with conventional classification algorithms. To resolve this problem the skewness in the dataset must be reduced.

Algorithms are designed that involves utilizing cost-sensitive learning techniques, such as adjusting the loss function on a per-sample basis by taking into account the ratio of the minority class. This entails penalizing samples that are difficult to classify, typically those belonging to minority classes, by assigning them higher loss values (Tang & He, 2017). In order to address the issue of class imbalance on image datasets, the samples with the minority class are augmented artificially using various techniques. These techniques entail altering already-existing images (Shorten & Khoshgoftaar, 2019). These adjustments cover a variety of techniques, such as brightness or contrast adjustments for color representations and geometric transformations like flipping, cropping, rotation, translation, and noise injection. In addition, methods like adding or removing parts from the images or using kernel filters like Gaussian filtering are used (Zhong et al., 2020; Escobar Díaz Guerrero et al., 2024; Schaudt et al., 2023).

Hagos et al. (2021) introduced a method involving the generation of a matrix associated with weights, where each element holds a specific value. This weighted matrix is then utilized within the weighted dice overlap loss function to address the challenges posed by class imbalance (Hagos et al., 2021). In imbalanced datasets, one or more classes with dominant dataset results in the overfitting. This imbalance results in the deep learning models to detect the class with the majority class of the labels as their detection leads to higher scores. Lin et al. (2017a) proposed a focal loss function that uses cross-entropy (CE) to reduce the impact of the majority class and helps to classify the samples of the minority class. This loss function helps to overcome the problem of class imbalance allows the samples to classify with better accuracy.

Further to resolve the problem of class imbalance for the chore of classification in the images, this article proposes an under-sampling strategy based on reducing the samples of majority class leveraging the unsupervised clustering for partitioning the majority class into clusters within the datasets.

Motivation: Traditional classification methods can struggle to accurately identify minority classes due to their limited representation in the dataset. The proposed technique, Majority Clustering for Imbalanced Image Classification (MCIIC) revolutionizes traditional binary classification problem by seamlessly transforming them into multi-class challenges, thereby broadening the scope for comprehensive classification solutions. By utilizing the elbow method, we determine the optimal number of clusters for the majority class and assign each cluster a new label. This allows to reduce the deviation between majority and the minority class samples and helps in solving the problem of class imbalance in image datasets. In this work, datasets like COVID/no-COVID, bald/no-bald, tuberculosis/normal are used where the class of interest is represented by rare samples which can be consider as outliers rather than simple normal class. The challenge is not just balancing the class but proper classification of those rare samples from the dataset using the cost-effective techniques.

The article includes the following contributions:

1. An efficient pre-processing under-sampling method MCIIC, is proposed to reduce the problem of class imbalance on image datasets.

2. The proposed approach integrates clustering to improve the classification with in image datasets.

3. The proposed approach is integrated with weighted augmentation method and uses under sampling along with oversampling to classify the samples of the image.

4. Integration of MCIIC along with weighted classification is done to prove the efficiency of the proposed approach over the naïve methodologies.

The remainder of the article is structured as follows: ‘Related Work’ delves into the background of the class imbalance problem, ‘Proposed Model’ outlines the proposed model, and ‘Experiments and Results’ presents the experimental results obtained and ‘Conclusion’ concludes the article with the future prospectives.

Related Work

In binary classification problem, data can be divided into two classes. However, in some binary classification, the scenarios of class imbalance can occur. This can lead to poorly trained models when the focus of the model is the minority class as compared to majority class (Wei et al., 2013; Herland, Khoshgoftaar & Bauder, 2018). An instance of class imbalance in the field of medical science is in the detection of COVID, where it is important to detect the presence of covid infection in a patient, therefore stopping the further spread of infection. The conventional algorithms often exhibit a tendency to over-classify the majority class because of their higher prior probability. Consequently, instances belonging to the minority class are frequently misclassified in comparison to those of the majority class. To address this issue, various resampling techniques have been employed in the past. These techniques include over-sampling, under-sampling, and hybrid methods that combine both approaches. The over-sampling method involves reproducing of the data points in the minority class to alter the diffusion of the data points in that class. These samples are randomly replicated based on the existing samples in the class using well defined procedures (Douzas & Bacao, 2018; Kovacs, 2019). On the other hand, under-sampling is a non-heuristic approach that randomly discard the samples from the dominant class to solve the problem of class imbalance (Sun et al., 2023; Lin et al., 2017b). The weakness of this method is that it can eliminate the data that can be potential useful for the classification process. Another disadvantage of this strategy is that the random sample from the majority class cannot be selected from the under-sampled class. To overcome the problem associated with over-sampling and under-sampling method, the hybrid sampling methods were introduced that incorporates combination of over-sampling and under-sampling methods. These hybrid methods are either resampling methods combined with ensemble methods or ensemble methods combined with resampling methods (Han et al., 2019). The literature emphasizes the significance of data resampling in mitigating the challenges posed by class imbalance (Provost, 2000).

In image datasets, the problem of class imbalance is mitigated through augmentation of data (Achicanoy, Chaves & Trujillo, 2021). Traditionally, data is augmented by performing different graphical transformation on the images including rotation, translation, scaling, blurring, illumination adjustments, etc. However, these transformations at the image level often fail to adequately enhance the distinct separation between different classes, particularly when this separation relies on higher-level features (Theodoropoulos et al., 2024). To overcome this, generative adversarial networks (GAN) were used to generate the data to improve the process of classification. However, large set of training datasets are required to train these GAN models and it is difficult in case of unrepresented data to train the model as per the minority class samples. To overcome this, different types of GANs are proposed with auto-encoder mechanisms, conditional GAN, cost-sensitive GAN, augmentation GAN and many more (Huang & Jafari, 2023). Liu, Mo & Zhong (2023) introduced the concept of federating learning using semi-supervised approach based on a combination of pseudo-label construction and regularization for classifying imbalanced medical images.

On the contrary, under sampling is utilized to achieve class balance by reducing samples from the majority classes or dividing the majority class into multiple subsets. Integrating clustering techniques with under sampling proves effective in managing imbalanced datasets dominated by majority classes. Krawczyk, Bellinger & Corizzo (2021) introduced CUS, a clustering-based under sampling method, where majority class instances are clustered and under sampled based on their information content to create balanced datasets. Another under sampling method, LOFCUS, introduced by Yuanyuan & Liyong (2018), focuses on class overlap. LOFCUS employs a local outlier factor (LOF) and boxplot to detect and eliminate noisy samples from the training set, while retaining samples crucial for classification based on class overlap. Through this approach, LOFCUS aims to maintain the original data distribution while enhancing classifier accuracy (Yuanyuan & Liyong, 2018). However, achieving improved class balance remains challenging with the LOFCUS method.

Based on the literature, multi-class imbalance problem can lead to define the problem in better way as most of the algorithms are developed for binary class classification problem. It can lead to solve the problem of class imbalance more effectively with less complexity. There is also the problem to know about the number of classes in which majority class can be divided to generate multiple classes. To overcome the aforementioned problems, the MCIIC algorithm is proposed.

Proposed Model

In this section, the proposed model, Majority clustering for Imbalance Image classification (MCIIC) is discussed in detail. The proposed model consists of the following steps as shown in Fig. 1.

Input dataset

The proposed model, MCIIC, is used to solve the problem of classification where the dataset is class imbalanced. The number of instances belonging to one class greatly outnumbers the instances in another class. The dataset has binary classes, and the occurrence of one class is much less frequent than others. Several machine learning algorithms exhibit a bias towards the majority class, resulting in suboptimal predictive performance for the minority class. To deal with this problem, data need to be pre-processed before classifying. In MCIIC, the dataset used are of large size, so first stage involves extraction of features.

Feature extraction

Feature extraction plays important role for effectively handling large image datasets by reducing dimensionality, helps to enhance the generalization of the image. To perform this task, the model uses Residual Network (ResNet-18). The key advancement in ResNet is the introduction of Residual blocks. ResNet is a combination of traditional CNNs with residual blocks, where the residual blocks allow the network to extract more advanced and hierarchal features. These residual connections allow the model to propagate gradients effectively through deep layers, allowing the learning of complex features. The ResNet-18 model significantly decreases the complexity of the image by reducing its large dimensions to just 512 features. The ResNet model is pre-trained on an ImageNet dataset and can classify images into one thousand object categories. Figure 2 shows the layered structure of a normal ResNet-18 model (Escobar Díaz Guerrero et al., 2024).

Figure 1 A systematic approach outlining the proposed methodology.

Figure 2 Architecture of ResNet-18 model.

In the proposed model a ResNet-18 model (originally trained on ImageNet data set) is used as base. The ResNet-18 model comprises of eighteen layers, where the first 17 layers are used for feature extraction and final layer is responsible for predicting the image label. These initial layers are crucial for extracting significant features (512 features) from the input images. These extracted features are stored in an array. Subsequently, k-means clustering is used to divide majority class into multiple classes based on the extracted features.

Elbow method

In the proposed model, to address class imbalance in the data, the majority class is partitioned into multiple clusters using k-means clustering. The optimal number of clusters (k) is determined using the elbow method. This approach requires minimizing a sum of squared errors (SSE): (1) SSE= ∑k=1cluster_n ∑xj∈Xxi−ck

where k = Nc is the optimal number of clusters where the elbow is formed in the graph, xi is pixel value of an image in the dataset and ck is the cluster centre which is initialized manually and cluster_n is the maximum number of clusters to be considered. The elbow method helps to finalize the value of cluster_n for k-means clustering where the elbow point in Fig. 3 points to the optimal number of clusters to considered for the process of clustering.

The algorithm for the elbow method is given below Algorithm 1:

Algorithm 1 Algorithm for elbow method.

Input:	
- Data: Majority samples for clustering	
- cluster_n: Maximum number of clusters to consider	
Output:	
- Optimal number of clusters (k)	
1. Initialize an empty list to store the SSE values for different values of k.	
2. For each value of k from 1 to cluster_n:	
a. Perform k-means clustering on the data, using k clusters.	
b. Calculate the total within-cluster sum of squared error (SSE) for the clustering.	
c. Append the SSE value to the list.	
3. Plot a curve between the values of k (x-axis) and the corresponding SSE values (y-axis).	
4. Identify the “elbow point” on the plot, where the rate of decrease in SSE slows down significantly.	
5. The value of k at the elbow point is assumed as the optimal number of clusters.	
6. Return the optimal number of clusters (k).	

Figure 3 shows the graph of elbow method for COVID/no-COVID test dataset providing the optimal value of k as the output. The below given graph forms a curve at k = 5. The obtained value of k is the number of classes in which the majority class will be divided using k-means clustering.

Figure 3 Graph of elbow method for COVID/no-COVID dataset.

According to the elbow method, the determined number of clusters is applied to partition the majority class. The dataset M1 representing the majority class is segmented into Nc clusters using k-means clustering, with each cluster assigned a new class label. This procedure results in a reduction of the number of images per class as the number of majority classes increases to Nc. Consequently, this transforms the original binary-class problem into a multi-class problem with Nc+1 total classes, where the (Nc+1)th class represents the minority class.

Training the model using transfer learning

Once the data is classified into Nc+1 classes, it can be used to train the required model. These models can be trained from scratch i.e., training a neural network without relying on pre-existing knowledge or pre-trained weights. Alternatively, there is an option of employing machine learning techniques such as transfer learning to train new models based on previously trained models, particularly useful when dealing with smaller datasets.

Transfer learning is a concept of machine learning which involves using a pretrained neural network model to improve the learning process on another task, i.e., model trained on one task is adapted for second task (Shorten & Khoshgoftaar, 2019). The process of transfer learning involves selecting a pretrained model (ResNet-18 model, trained on a large data set), removing the ending layers responsible for specific tasks, and finally introducing new layers after the original layers. These layers are responsible for training the model for new tasks. The last step involves training the model for the new task using new data sets.

Transfer learning is often considered better than training a model from scratch because it provides data efficiency, faster convergence, and improved generalization. In many real-world scenarios, obtaining a large labelled dataset for a specific task can be challenging and expensive. By leveraging a pre-trained model’s learned features from a related task, transfer learning enables the adaptation to a target task even with limited labelled data. As the pre-trained model has already learnt generic features from a different but related task, transfer learning provides quicker convergence and reduces resource requirements. Figure 4 shows the difference between training a model from scratch vs transfer learning as learned knowledge is transferred while training the new model resulting into improved efficiency (Zhong et al., 2020).

Figure 4 (A) Building the training model from scratch. (B) Training the model using Transfer learning.

Thus, the goal of using transfer learning for training the model is to leverage a pretrained model’s features without extensively retraining the complete model. The dataset is split into 80-10-10 ratio for training, validation and testing. Training is now performed on the new multiclass dataset using transfer learning. ResNet-18 pretrained on ImageNet dataset, is used for training the dataset to achieve the desired outcomes.

Validating the training model and tuning of the hyper-parameters

During the training process, the model’s weights are continuously updated to minimize a chosen loss function (e.g., cross-entropy loss) on the training data. However, to prevent overfitting and ensure the model generalizes well to unseen data, the model’s performance on the validation set is monitored. The best model weights are those that result in the highest accuracy (or lowest loss) on the validation set and with the less computational cost. The standardized parameters are chosen for the process of initialization to produce more optimized results.

Analysing the performance

Once the model gives output for multiclass, this is then converted to binary class to obtain the final classification. To test the accuracy of the model, the multi-class data is transformed into binary class data using the process of mapping. The mapping is based on the labels, such that classes from the majority class are mapped into one class and label with the minority class are mapped into desired class. The performance of the model is evaluated on a binary validation dataset by mapping its predictions to binary values, and selecting the best model weights based on the highest accuracy achieved on the validation set during training.

The architecture of the proposed methodology for the classification of imbalanced image dataset is shown in Fig. 5. The input dataset is imbalanced dataset of tuberculosis images and normal images, and the proposed methodology aims to classify minority class images (tuberculosis images) correctly.

Figure 5 An architecture for proposed majority clustering for imbalanced image classification.

Analysing the time-complexity

To determine the time-complexity of the proposed method assuming the following parameters:

M1-Represents images in the majority cluster

Nc-Represents number of clusters in which majority class is divided

d- Dimension of the dataset

L-Number of layers

P-Number of parameters

G-Number of hyper-parameter combination

T-Number of training runs

K-Number of time k-means clustering algorithm run for different values of Nc

Performing the elbow method using k-means clustering takes O(M1NcKd). The classification process using the model trained with transfer learning takes O(M1LP) and hyper-parameters tuning requires O(GT). The overall time-complexity of the proposed algorithm takes O(M1NcKd+M1LP+GT).

Experiments and Results

This section discusses about the datasets used for experimentation, evaluation metrics and various comparison techniques used to assess the performance of the proposed model. The proposed approach is implemented using following computing infrastructure:

Computing infrastructure:

OS: Ubuntu 22.04.4 LTS

CPU: Intel(R) Core(TM) i5-6600

GPU: NVIDIA GeForce RTX 3090

RAM: 32 GB

To explore the efficiency of the proposed model MCIIC, the proposed model is further integrated with weighted data augmentation and weighted data classification.

(a) The first approach involves conducting the analysis without any additional changes to the dataset. This means that no adjustments are made to the dataset before the training and testing of the model. All the data points are used without any modifications. This provides a baseline for understanding how the model performs using the given data without any additional processing. This allows for the evaluation of model’s raw performance on the dataset without any adjustments.

(b) The second approach incorporates the proposed algorithm MCIIC with the weighted classification into the evaluation process to assess its impact on the results. Since most machine learning algorithms are not good with biased classes, the concept of weights is applied by giving weights to both majority class and minority class (Schaudt et al., 2023). This difference in weights will influence the classification during training process. It reduces the misclassification made by minority class by setting a higher-class weight and simultaneously reducing the weights of the majority class. Here weighted loss function is used instead of normal weight function to train the model. The weights are used to assign the higher penalty while miss-classification of the minority class. This will increase the cost of miss-classification of the minority class.

(c) In the third approach, the MCIIC is integrated with weighted augmentation. In weighted augmentation the weights are assigned to the classes based on the number of the samples in the specific class. The minority class is provided with higher weights and majority class with lower weights. The class with the higher weights is having high probability of the data needs to be augmented. To solve the problem of class imbalance, the minority class is biased with higher weights to pay more attention for the augmentation. For example, if class a, b and c have 10, 20 and 30 samples each, respectively, then augmentation probability of class a will be 1, class b will be 0.5 and class c will be 0.33. Different types of image augmentation techniques include image rotation, image shifting, image blurring, image flipping, and image noising. These techniques help in efficient training of model, even in the absence of a large dataset.

For training and testing the model using the described approaches, various datasets are required. These datasets are explained below.

Datasets

1. Bald/no-bald dataset: This dataset contains the images of faces. It is divided into two types of images: one with the faces of bald individuals and other with the faces of individuals with hair on their scalp. This dataset is acquired from CelebA dataset (Li, 2018). CelebA dataset has 40 different classes, where bald/no-bald attribute category stands out having the most notable difference in the number of samples of these two classes. The size of the training data includes number of bald images are 158,442 and number of no-bald images are 3,637. The imbalance ratio is 2.3%. Some of the images belonging to the “bald” and “no-bald” classes are shown in Figs. 6 and 7.

2. Normal/tuberculosis dataset: The second dataset consists of medical chest X-ray images. One category, named as “Normal” consists of healthy chest images, while the other category named “Tuberculosis”, consists of X-rays depicting chest images of people infected with tuberculosis. These images have been obtained from the imbalanced Tuberculosis and Pneumonia dataset (Roshan, 2022). For binary classification, only the “normal” and “tuberculosis” classes are considered. The size of the training data includes number of normal images without tuberculosis are 7,430 and number of images with tuberculosis are 1,510. The imbalance ratio is 20.3%. Some of the images belonging to the “Normal” and “Tuberculosis” classes are shown in Figs. 8 and 9.

3. COVID/no-COVID dataset: This dataset includes chest X-ray images used for covid detection. One category, named “COVID” consists of chest X-ray images showing signs of COVID infection, while the “no-COVID” category consists of images without any COVID infection. These images are obtained from the COVIDx CRX-3 dataset (Roshan, 2022). To introduces class imbalance, approximately 20% of the images from the COVID class are randomly selected. The size of the training data includes number of no-COVID images are 11,352 and number of COVID images are 681. The imbalance ratio is 6%. Some of the images belonging to the “COVID” and “no-COVID” classes are shown in Figs. 10 and 11. Table 1 shows the summary of the features of the datasets used for evaluation.

Evaluation metrics

To assess the effectiveness of the proposed algorithm compared to other classification techniques, confusion matrices are employed to analyze how well the predictions align with the actual ground truth values. When making predictions, instances from the minority class are rare occurrences where a positive outcome is indicated. In such cases, the majority class is typically considered as negative. The metrics used to validate the performance of the proposed algorithms are:

Figure 6 Images in the “bald” class.

Figure 7 Images in the “no-bald” class.

Figure 8 Images in the “normal” class.

Accuracy: In a classification model, accuracy measures the correctly predicted labels, indicating the proportion of correctly predicted cases out of all instances. It is calculated as Eq. (2).

(2) Accuracy=TP+TNTP+TN+FP+FN

where TP (true positive) denotes correctly predicted positive cases, TN (true negative) denotes correctly predicted negative cases, FP (false positive) indicates incorrectly predicted positive cases, and FN (false negative) indicates incorrectly predicted negative cases.

Precision: It evaluates the precision of positive predictions, indicating how reliably the model identifies positive instances correctly. The better value of high precision indicates the low value of incorrectly predicted positive instances. Equation (3) represents the measure of the precision as (3) Precision=TPTP+FP.

Recall: The recall metric evaluates how well the model can correctly identify positive instances among all the actual positive instances. Equation (4) represents the measure of the recall as (4) Recall=TPTP+FN.

The higher vale of recall implies low value of the false negative instances predicted by the model.

F1_score: This measure balances the precision and recall when both are equally important. It is calculated as the harmonic mean of the precision and recall as Eq. (5) (5) F1score=2∗Precision∗RecallPrecision+Recall.

Furthermore, in our research, the confusion matrix plot proves highly valuable due to its insensitivity to sample balance, making it particularly adept at handling skewed data distributions. Moreover, these plots demonstrate greater responsiveness to variations in model performance, with higher values indicating a greater likelihood that the classifier will effectively rank positive samples, thereby enhancing classification accuracy (Yuan et al., 2021).

To choose the consistency of the experimentation, the training parameters and configuration chosen are as follows (Box 1):

BOX 1 Training Parameters

1. Preprocessing:

Training data

1. pytorch random-resize-crop with image size=224

2. image normalization using imagenet mean and standard deviation. ([0.485, 0.456, 0.406], [0.229, 0.224, 0.225])

Validation data

1. pytorch resize to img size=256, then center crop to img size=224

2. image normalization using imagenet mean and standard deviation. ([0.485, 0.456, 0.406], [0.229, 0.224, 0.225])

2. Model:

1. resnet18 pretrained on imagenet dataset

2. Total epochs: 50

3. cross entropy loss

4. optimizer: stochastic gradient descent with momentum=0.9, weight-decay=5e−4

5. batch-size=256

6. initial learning rate=0.005, reduced by 1/10th after every 12 epochs

3. Weighted augmentation

1. weight_for_class_i = total_samples / (num_samples_in_class_i * num_classes)

2. augmentations: random brightness change(0.75−1.25), random rotation(-10 to +10 degree), random horizontal flip

4. Weighted loss

1. weight_for_class_i = total_samples / (num_samples_in_class_i * num_classes)

Results and Discussions

To evaluate the performance of the proposed model, analysis is done with different integrated models considering the class imbalance problem. The classification is done using simple baseline model using ResNet-18, ResNet-18 with weighted augmentation, ResNet-18 with weighted classification, ResNet-18 with MCIIC, ResNet-18 with MCIIC and weighted augmentation, and ResNet-18 with weighted classification. After a direct comparison with the baseline method, and other modified models it is observed from Table 2. That classification results for bald/no-bald dataset where MCIIC model shows better accuracy as compared to other models. When MCIIC is applied with weighted augmentation is shows even better results.

Figure 9 Images in the “tuberculosis” class.

Figure 10 Images in the “COVID” class.

Figure 11 Images in the “No-COVID” class.

Table 3 shows the results of classification performed on normal or tuberculosis dataset. It shows that classification performed using MCIIC with weighted augmentation outperforms in accuracy.

Recall and F1-score are better when the classification is performed using MCIIC with weighted classification as compared to base model of ResNet-18 and other hybrid models with ResNet-18.

Table 4 depict the results of comparison between ResNet model and the proposed MCIIC model for the COVID/no-COVID dataset, respectively. It can be observed form the results that proposed model outperforms in accuracy, precision and F1-score while hybrid model of MCIIC with weighted classification performs well giving better recall values.

Figure 12 shows the confusion matrix plots for bald/no-bald dataset. Figure 12A shows the classification performed using normal ResNet-18 model, Fig. 12B shows classification performed using proposed MCIIC, where the number of samples in the minority class (bald samples) has increased for all three classes with the increase in the value of true positives and the number of majority class samples (no-bald samples) has reduced as false negative samples are reduced. Figure 12C shows classification performed using MCIIC integrated with weighted augmentation and Fig. 12D shows classification performed using MCIIC integrated with weighted classification. The proposed method outperforms when integrated with weighted augmentation giving better results on imbalanced datasets. Figure 13 shows the confusion matrix plots of the COVID/no-COVID dataset. Here Fig. 13D shows the best results where classification accuracy of minority class increases with the proposed approach integrated with weighted classification. Figure 14 shows the confusion matrix plots for tuberculosis/normal dataset. Here the proposed MCIIC results shown in Fig. 14B achieves maximum accuracy.

Table 1 Characteristics of the datasets.

Datasets	Attributes	No. of samples in majority class	No. of samples in minority class	Imbalance ratio	
CelebA dataset (Li, 2018)	Bald/No-Bald	1,58,442	3,637	2.3%	
Pneumonia dataset (Roshan, 2022)	Normal/Tuberculosis	7,430	1,510	20.3%	
COVIDx CRX-3 dataset (Roshan, 2022)	Covid/No-Covid	11,352	681	6%	

Table 2 Test data results for bald/no-bald dataset.

Method	Accuracy	Precision (Minority class)	Recall (Minority class)	F1-score (Minority class)	
ResNet-18	0.944	0.842	0.820	0.831	
ResNet-18 + weighted-aug	0.940	0.901	0.725	0.804	
ResNet-18 + weighted-clf	0.937	0.777	0.884	0.827	
ResNet-18 + MCIIC	0.953	0.866	0.852	0.859	
ResNet-18 + weighted-aug + MCIIC	0.954	0.921	0.799	0.856	
ResNet-18 + weighted-clf +MCIIC	0.943	0.811	0.862	0.836	
Notes.

Bold values represent the algorithms with the highest accuracy, precision, recall, and F1-score, highlighting the performance of the proposed technique.

Table 3 Test data results for normal/tuberculosis dataset.

Method	Accuracy	Precision (Minority class)	Recall (Minority class)	F1-score (Minority class)	
ResNet-18	0.972	0.820	0.658	0.681	
ResNet-18 + weighted-aug	0.971	0.723	0.644	0.681	
ResNet-18 + weighted-clf	0.964	0.592	0.863	0.693	
ResNet-18 + MCIIC	0.973	0.822	0.685	0.746	
ResNet-18 + weighted-aug + MCIIC	0.977	0.731	0.671	0.700	
ResNet-18 + weighted-clf +MCIIC	0.975	0.696	0.877	0.776	
Notes.

Bold values represent the algorithms with the highest accuracy, precision, recall, and F1-score, highlighting the performance of the proposed technique.

Table 4 Test data results for COVID/no-COVID dataset.

Method	Accuracy	Precision (Minority class)	Recall (Minority class)	F1-score (Minority class)	
ResNet-18	0.964	0.737	0.554	0.632	
ResNet-18 + weighted-aug	0.981	0.691	0.275	0.393	
ResNet-18 + weighted-clf	0.976	0.480	0.826	0.607	
ResNet-18 + MCIIC	0.988	0.755	0.657	0.703	
ResNet-18 + weighted-aug + MCIIC	0.984	0.680	0.536	0.600	
ResNet-18 + weighted-clf +MCIIC	0.974	0.462	0.921	0.616	
Notes.

Bold values represent the algorithms with the highest accuracy, precision, recall, and F1-score, highlighting the performance of the proposed technique.

Figure 12 Confusion matrix plots of bald/no-bald data.

(A) Normal classification. (B) Proposed MCIIC (C) MCIIC with weighted augmentation. (D) MCIIC with weighted classification.

Figure 13 Confusion matrix plots of COVID/no-COVID data.

(A) Normal classification. (B) Proposed MCIIC. (C) MCIIC with weighted augmentation. (D) MCIIC with weighted classification.

Figure 14 Confusion matrix plots of normal-tuberculosis data.

(A) Normal classification. (B) Proposed MCIIC. (C) MCIIC with weighted augmentation. (D) MCIIC with weighted classification.

Upon analysing the outcomes of the proposed model against those of the baseline method, it becomes evident that the proposed model outperforms the baseline across all approaches. The results prove the effectiveness of the model for the process of image classification even with skewed datasets. Therefore, proving its efficiency and scope in various applications.

Conclusion

In conclusion, this study introduces a hybrid approach, Majority Clustering for Imbalanced Image Classification (MCIIC), tailored to address challenges inherent in classifying imbalanced datasets. By leveraging the elbow method in conjunction with k-means clustering, the method clusters the majority class into n subgroups, thereby transforming the binary classification task into a multi-class problem encompassing (n+1) comparably balanced classes. This novel framework was rigorously evaluated across three benchmark datasets: COVID/no-COVID images, bald/no-bald images, and tuberculosis/no-tuberculosis images. To further enhance model performance, we integrated weighted augmentation and weighted classification techniques. The results demonstrate significant performance gains, achieving an impressive accuracy of 97.7%. This underscores the effectiveness of our approach in mitigating class imbalance and advancing state-of-the-art methodologies for image classification tasks. The limitation of this research is that it is not tested for multiclass datasets. Optimal clusters are calculated using the elbow method, which is computationally expansive. The limitation of the proposed methodology is that the performance of the MCIIC method may vary across different datasets and applications. Its effectiveness may not generalize well to all scenarios, particularly those with highly diverse or complex class structures. It is highly sensitive to clustering used for grouping the majority class to multi-class problem.

Further, the proposed methodology could be extended to multi-class imbalanced classification problems with the ability to handle the problem of class imbalance in the datasets with the multiple classes. Different alternative clustering techniques can be explored for automatically selecting the number of clusters that reduce computational cost and helps to optimize the classification results.

Supplemental Information

Supplemental Information 1 Code for Experiment

Supplemental Information 2 Python code for model training

Supplemental Information 3 Python code for database loading

Supplemental Information 4 Python code for model creation

Supplemental Information 5 Python code for model evaluation

Supplemental Information 6 Datasets and Hardware Used

Supplemental Information 7 Motivation

Supplemental Information 8 Readme

Supplemental Information 9 Description of Model

Supplemental Information 10 Limitation

Additional Information and Declarations

Competing Interests

Author Contributions

Data Availability

The authors declare there are no competing interests.

Keshav Sharma conceived and designed the experiments, performed the experiments, performed the computation work, prepared figures and/or tables, and approved the final draft.

Jyoti Arora conceived and designed the experiments, performed the experiments, analyzed the data, authored or reviewed drafts of the article, and approved the final draft.

Pooja Kherwa analyzed the data, prepared figures and/or tables, and approved the final draft.

Zainab Alansari performed the computation work, authored or reviewed drafts of the article, and approved the final draft.

The following information was supplied regarding data availability:

The Imbalanced Tuberculosis and Pneumonia dataset is available at Kaggle: https://www.kaggle.com/datasets/roshanmaur/imbalanced-tuberculosis-and-pnuemonia-dataset.

The code is available at Zenodo. sharma, . keshav . (2025). MCIIC Code. Zenodo. https://doi.org/10.5281/zenodo.15049576.

The CelebFaces Attributes (CelebA) Dataset is available at Kaggle: https://www.kaggle.com/datasets/jessicali9530/celeba-dataset.

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
