# Peer review of "Majority clustering for imbalanced image classification"

_PeerJ Computer Science, doi:10.7717/peerj-cs.2891_

## Round 0.1 · original submission · Major Revisions

Dear authors: the reviewers made a great job and provide you with a detailed plan for improvements. Please, address all critical comments and consider suggestion of the proposed changes and expansions.

Reviewer 1 ·

Basic reporting

The study designed a method for imbalanced data clustering. Please consider the following comments.
1. The related work is not well organized. Imbalanced classification can be organized into three main categories.

2. In line 186, how to set k for k-means?
3. The authors did not clearly clarify how transfer learning work.
4. How about the time complexity of the proposed method?

Experimental design

This is no statistically analysis.

Validity of the findings

Seems good.

Additional comments

No additional comments.

Cite this review as

Reviewer 2 ·

Basic reporting

The manuscript is written in fairly good English but requires some improvements and there are clear indications of parts being generated or influenced by Large Language Models (LLMs). Phrases like "this meticulous process" and "our findings underscore" are generic and stylistically characteristic of AI-generated text. These detract from the scientific tone and should be replaced with concise and precise language.
The introduction provides a general context for class imbalance challenges but fails to frame the problem adequately for the specific datasets used (e.g., bald/no-bald, normal/tuberculosis, covid/no-covid), which could be described as anomaly or outlier detection problems rather than general class imbalance tasks.
While the paper includes a related works section, it lacks depth and relevance. The cited works cover broad topics such as data augmentation and federated learning, which are tangential to the primary focus. The authors fail to address directly relevant literature, such as existing clustering-based methods for undersampling in imbalanced image classification. A simple Google Scholar search for "clustering for undersampling imbalanced image classification" yields numerous works that should/could have been cited, some of which could serve as benchmarks for comparison as well.
https://scholar.google.com/scholar?hl=en&as_sdt=0%2C10&q=clustering+for+undersampling+imbalanced+image+classification&btnG=
Points for Improvement:
1. Revise language to eliminate vague or overly generic expressions, ensuring precise and specific wording throughout the text.
2. Reframe the introduction to reflect the nature of the datasets and bring appropriate theoretical grounding.
3. Expand the related works section to include relevant clustering-based undersampling techniques and position the proposed method within this context.
4. Clarify the specific novelty of the proposed method compared to existing approaches in the clustering-for-undersampling domain.
5. Rephrase exaggerated statements (e.g., "revolutionizes traditional binary classification problems") to maintain an objective tone.
6. Provide additional clarification for technical terms like "weighted augmentation" for readers less familiar with the field.
7. Review grammar and syntax for minor inconsistencies, such as verb usage and sentence construction.
8. Improve figure and table captions to summarize findings more comprehensively.

Experimental design

The manuscript shows an experimental design but falls short in providing critical details about the datasets used, including an explanation of why they are chosen. I would also say that three data sets are not enough to validate the paper’s claims. The datasets (bald/no-bald, normal/tuberculosis, and covid/no-covid) are mentioned briefly, but essential descriptive statistics and contextual details are missing. For a robust evaluation, the following aspects need to be addressed:
1. Dataset Description:
The paper does not provide sufficient information about the datasets, such as the number of samples in each class, the number and description of attributes, and their relevance to the problem domain. For example:
o What is the total number of images in each dataset?
o How are the images distributed between the majority and minority classes?
o Are there any inherent biases in the datasets that might influence the results?
2. Contextual Relevance:
While the datasets are presented as examples of imbalanced classification problems, they may be more representative of anomaly detection tasks. The authors need to justify their selection of these datasets and explain why they are appropriate for evaluating the proposed method.
3. Benchmarking Datasets:
The manuscript could strengthen its experimental design by including a broader range of datasets, especially ones used in prior clustering-based undersampling studies. This would enable a more comprehensive assessment of the method's performance and generalizability.
4. Data Preprocessing and Augmentation:
While some preprocessing details are provided (e.g., feature extraction with ResNet-18), the paper lacks clarity on how the datasets were prepared for clustering. For example:
o Were all images preprocessed in the same way?
o What resolution and normalization techniques were used?
Points for Improvement:
1. Provide detailed descriptions of all datasets, including:
1. Number of samples for each class (majority/minority).
2. Number and types of attributes (e.g., image resolution, file format).
3. Potential biases or limitations.
2. Justify the selection of datasets and their relevance to the problem domain.
3. Include a table summarizing the key characteristics of the datasets for better readability.
4. Consider using additional datasets from prior clustering-based studies to strengthen the benchmarking and generalizability of the results.
5. Expand on preprocessing steps, including attribute transformations, normalization, and any augmentation techniques applied.
6. Discuss the impact of hyperparameter tuning in the model training process, particularly its effect on reproducibility and performance.

Validity of the findings

The experimental results show that MCIIC outperforms the baseline models on the selected datasets. However, the choice of benchmarks and datasets limits the validity and generalizability of these findings. The datasets used are highly specific and not representative of broader imbalanced classification problems. Without proper benchmarking against relevant clustering-based undersampling methods, the conclusions drawn are incomplete and lack sufficient support.
The conclusions align with the results presented but are overly optimistic. While the authors claim significant improvements, they fail to discuss the limitations of the approach, such as its computational cost, potential sensitivity to hyperparameters (e.g., the number of clusters), and lack of testing on multiclass datasets. Furthermore, the paper does not adequately address the method's scalability or generalizability to larger datasets or other domains.
Points for Improvement:
1. Include benchmarks against existing clustering-based undersampling methods to validate the findings.
2. Discuss the limitations of the proposed method, including computational cost and sensitivity to hyperparameters.
3. Provide an in-depth analysis of scalability and generalizability to larger datasets or different types of imbalanced problems.
4. Reframe the conclusions to reflect these limitations and suggest realistic future directions for research.
5. Propose clearer future directions, such as adapting MCIIC for multiclass imbalanced datasets or integrating it with other state-of-the-art methods.
6. Discuss scenarios where MCIIC may not perform as well, providing a more balanced and nuanced view of its applicability.

Additional comments

. Clustering-based Approach Uniqueness:
While the manuscript presents clustering as a novel approach to address class imbalance, it could further clarify how this technique differentiates itself from similar clustering-based methods in the literature.
. Computational Complexity:
A detailed analysis of the computational complexity of the proposed methodology, particularly the clustering and transfer learning stages, would be beneficial. This is especially important for practitioners considering MCIIC for large-scale datasets.
. Model Comparison:
The comparisons with baseline and hybrid models are insightful, but additional benchmarks against alternative advanced methods (e.g., GAN-based oversampling, advanced ensemble methods) would provide a more comprehensive evaluation.
. Visual Interpretability of Clustering:
Include visualizations or interpretations of the clusters formed during the majority class partitioning process. For example, show how features are distributed within the clusters and how this impacts the classifier's decision-making process.
. Bias in Minority Classes:
While the study emphasizes handling class imbalance, it could also discuss how inherent biases within the minority class (e.g., noise or low-quality data) might affect the model’s overall performance and how MCIIC mitigates such issues.
. Scalability Concerns:
Although the datasets used are diverse, the paper does not explicitly address how the method scales to much larger datasets or domains requiring real-time predictions. Discussing scalability in terms of computational demands and storage requirements would strengthen the study’s practical implications.
. Future Work: The future directions mentioned in the paper are vague. The authors should propose concrete avenues for research, such as adapting MCIIC for multiclass imbalances or integrating it with anomaly detection frameworks.
. Terminology Consistency: Ensure consistent terminology throughout the manuscript. For example, "weighted augmentation" and similar terms should be clearly defined and consistently used.

Cite this review as

---

## Round 0.2 · accepted · Accept

Thank you for carefully following the comments and suggestions of the reviewers. You have improved your submission substantially.

Reviewer 2 ·

Basic reporting

• The manuscript is generally written in clear and professional English, with improvements over the previous version.
• The introduction provides adequate context for the study, clearly outlining the problem of class imbalance in image classification and motivating the need for the proposed method. The background and motivation are now more explicit, and the literature cited is relevant and up-to-date.
• The structure of the article largely conforms to PeerJ standards and discipline norms. The sections are logically ordered, and the flow from introduction to methods, results, and conclusion is coherent.
• The introduction successfully introduces the subject and motivation, making it clear what gap the paper intends to fill. The contributions are now listed more explicitly, which is helpful.

Suggestion for improvement:
• Further polish the language throughout the manuscript to ensure clarity for an international audience. Consider a professional language review.

Experimental design

• The article is within the aims and scope of the journal and presents an AI application relevant to image classification.
• The investigation is technically sound, and the experimental methodology is described in reasonable detail. The process of majority class clustering, the use of the elbow method, and the integration with weighted augmentation are now better explained.
• The methods section describes the datasets, models (including ResNet-18), and evaluation metrics (accuracy, precision, recall, F1-score). However, some details could be expanded for full reproducibility:
• Provide more explicit information about preprocessing steps, such as image normalization, resizing, and any augmentation parameters used.
• Clarify the clustering implementation (e.g., initialization, random seeds, and computational resources).
• Include a reproducibility checklist or script, or specify how readers can access the code and data.
• There is a discussion of data preprocessing, but it could be more detailed, especially regarding how minority and majority classes are handled and how clusters are assigned as new class labels.
• Evaluation methods and assessment metrics are adequately described and appropriate for the problem.
• Sources are generally well cited, but ensure that all claims—especially those about comparative methods and prior work—are supported by references.

Validity of the findings

• The conclusions are well stated and limited to the supporting results. The experimental results are clearly presented, and the impact of the proposed method is demonstrated on several datasets.
• The experiments and evaluations appear satisfactory, with appropriate metrics and comparative baselines. Statistical significance or confidence intervals are not mentioned—consider including these to strengthen the findings.
• The argument is well developed and supported by the results, meeting the goals set out in the introduction.
• The conclusion identifies some limitations and future directions, but this section could be expanded. For example, discuss potential weaknesses of the approach (e.g., sensitivity to cluster number selection, computational cost) and outline more concrete future research steps.

Additional comments

• The manuscript is much improved from the previous version, with clearer structure and better articulation of the research problem and contributions.
• Figures and tables are appropriate and contribute to the clarity of the results. Ensure all figures are high resolution and axes are labeled clearly.
• The raw data and supplemental files are provided, but ensure that all supplemental materials are well documented with descriptive metadata for future readers.
• Consider adding a brief summary table comparing the proposed method with baseline approaches, highlighting strengths and weaknesses.

Cite this review as